# Usability of Smartbands by the Elderly Population in the Context of Ambient Assisted Living Applications

**Luís Correia** [1], **Daniel Fuentes** [1], **José Ribeiro** [1], **Nuno Costa** [1], **Arsénio Reis** [2], **Carlos Rabadão** [1], **João Barroso** [2] and **António Pereira** [1,3,*]

1   Computer Science and Communication Research Centre, School of Technology and Management, Polytechnic Institute of Leiria, 2411-901 Leiria, Portugal; luis.correia@ipleiria.pt (L.C.); daniel.fuentes@ipleiria.pt (D.F.); jose.ribeiro@ipleiria.pt (J.R.); nuno.costa@ipleiria.pt (N.C.); carlos.rabadao@ipleiria.pt (C.R.)
2   INESC TEC, University of Trás-os-Montes e Alto Douro, Quinta de Prados, 5001-801 Vila Real, Portugal; ars@utad.pt (A.R.); jbarroso@utad.pt (J.B.)
3   INOV INESC Inovação, Institute of New Technologies, Leiria Office, Campus 2, Morro do Lena-Alto do Vieiro, Apartado 4163, 2411-901 Leiria, Portugal
*   Correspondence: apereira@ipleiria.pt

**Abstract:** Nowadays, the Portuguese population is aging at a fast pace. The situation is more severe in the interior regions of the country, where the rural areas have few people and have been constantly losing population; these are mostly elderly who, in some cases, live socially isolated. They are also often deprived of some types of social, health and technological services. One of the current challenges with respect to the elderly is that of improving the quality of life for those who still have some autonomy and live in their own residences so that they may continue living autonomously, while receiving the assistance of some exterior monitoring and supporting services. The Internet of Things (IoT) paradigm demonstrates great potential for creating technological solutions in this area as it aims to seamlessly integrate information technology with the daily lives of people. In this context, it is necessary to develop services that monitor the activity and health of the elderly in real time and alert caregivers or other family members in the case of an unusual event or behaviour. It is crucial that the technological system is able to collect data in a nonintrusive manner and without requiring much interaction with the elderly. Smartband devices are very good candidates for this purpose and, therefore, this work proposes assessing the level of acceptance of the usage of a smartbands by senior users in their daily activities. By using the definition of an architecture and the development of a prototype, it was possible to test the level of acceptance of smartbands by a sample of the elderly population—with surprising results from both the elderly and the caregivers—which constitutes an important contribution to the research field of Ambient Assisted Living (AAL). The evaluation showed that most users did not feel that the smartband was intrusive to their daily tasks and even considered using it in the future, while caregivers considered that the platform was very intuitive.

**Keywords:** accessibility and usability; smartbands; Internet of Things; monitoring the elderly; health and physical activity; ambient assisted living

## 1. Introduction

Demographic projections predict an accelerated aging of the world population. It is expected that in 2050, the population over 60 years old will double when compared to the existing one in the same age group from the year 2000 [1].

The Portuguese population follows the world trend and is also ageing at a fast pace, particularly in the interior regions of the territory and in the rural areas which are populated by few residents [2]. This lack of population is due to several reasons, including the migration of the active population to the big urban centres or to other countries, pursuing better life conditions, professional stability, access to services, etc. Consequently, the few remaining residents in the rural environments are mostly elderly people who often live

geographically and socially isolated from family, friends and even caregivers. In many cases they do not have easy access to some types of services, such as social, health and technological ones. This situation results in a tremendous demand and supply imbalance, with an elderly population in need of care services and a scarce offer of proper support services [3].

To a certain extent, what this particular elderly population lacks is a proper social support network, which in previous times was provided by other younger people, e.g., younger family, neighbors and friends. Currently, the elderly can live in their own homes if they are healthy and fully autonomous or they must opt for an institutionalisation solution if they have lost some autonomy or suffer from a disease. There is not much room for a middle term solution.

Supporting life quality improvements can provide a solution for these people, though it is a challenge for the technology and the care services. The objective is for those who still have some autonomy to continue to live in their own homes instead of being moved to day care centres, nursing home or other social responses, which will move the elderly away from their own homes. This solution might be achieved by introducing unintrusive monitoring and care services in terms of health, social and well-being, safety and physical integrity. Technology should enable care services to be seamlessly provided as required. Moreover, it is necessary to consider the high level of technological illiteracy that results from the physical and cognitive limitations associated with the normative aging process. These are important factors to take into account when developing applications for this target audience [4].

The Internet of Things (IoT) paradigm has great potential to be used as a tool to support such unintrusive services, as one of its goals is to integrate information technology with the physical resources that people use in their daily lives [5]. This paradigm allows millions of physical agents to communicate thought the Internet. These objects can be of various types, e.g., buildings, vehicles, people, animals or even plants, which can transmit information about the surrounding environment or about their own state in an intelligent manner and without human intervention by using an Internet connection [6]. Due to the increasing use of the Internet for various purposes, the number of connected devices has increased exponentially. According to Cisco [7], this number will reach 500 billion by 2030.

Therefore, it is possible for this technological paradigm to create services that monitor the activity and health status of the elderly in real time and to alert their caregivers and/or family in the case of an atypical event or behaviour, hence realising the view of the Ambient Assisted Living [8]. In this fashion, the caregiver and/or family have up-to-date information about their well-being even if they are not in physical proximity relative to the elderly person.

For privacy reasons as well as due to the low technological literacy of the senior population, it is crucial that the system collects only the necessary data in a non-intrusive manner without requiring the elderly's active participation [9].

As the data related to the vital signs of the elderly are important information to monitor and the smartbands wearable devices are excellent candidates to collect this data, this study aims to assess the level of acceptance of the use of a smartband by senior users in their daily activities.

This work aims to answer the following research question: "What is the level of acceptance, by the elderly population, of the use of smartbands to monitor their daily activities?"

In order to answer this question, the following objectives were defined: analysis of monitoring systems for the elderly population; proposal of an architecture using the IoT paradigm in the context of AAL systems; implementation of a prototype; tests with elderly people and caregivers in a real environment; questionnaires to the participants in the test session and analysis of the results obtained. After all the objectives had been put into practice, it was possible to observe that the elderly people accepted the use of smartbands and the smartband became an object that they would consider using in their daily lives. On

the other hand, the caregivers considered the platform intuitive and that, in the future, they could use this service to monitor the elderly, which constitutes an important contribution to the research field of the AAL.

## 2. Related Work

There are several works focused to the real-time collection of biometric data in humans, such as walking distance, calories burned, heart rate, sleep quality, fall detection, etc. In this section we address other works related to this subject that are segmented into three groups: acceptance and usability of wearable devices; monitoring and support; recognition of activities.

### 2.1. Wearable Devices Acceptance and Usability

In the work presented in [10] and with respect to the acceptance and adoption of assistive electronic systems by the elderly, the conclusions suggested that the elderly population is willing to attempt to use such systems as long as they provide tangible benefits, even if they may not be aware of the latest technological developments in this field.

Some research work has been conducted in the field of the usability of wearable devices. In the work proposed in [11], the authors developed a study in which they analyse the usability issues of wearable devices, specifically smartwatches. In order to carry out this study, the authors recruited thirty users to wear this type of device, aged between 20 and 43 years and each user was followed for a week. These users would perform several daily tasks using their smartwatches. In some tasks the users had to interact with the device itself. The authors identified some usability problems associated with these tasks, including discomfort and localized pain caused by the smartwatches that are due in part to the size and weight of the device.

The authors of [12] designed a study with the aim of examining the reasons for the use of technologies that permit physical activity to be monitored, such as smartphones, smartwatches and tablets. A total of 1013 people that were aged over 50 years were interviewed. From the data collected, the authors concluded that 20.5% of the participants used devices to monitor their physical activity. Men, especially younger men, have a greater interest in the new technologies and are more likely to use mobile devices for physical activity monitoring. The authors identified the need for further studies on the motivational and usability aspects of health monitoring devices by older users.

The study presented in [13] indicates that social influences are facilitating conditions for the use of these type of devices. The Xiaomi Mi Band 2 smartband was used as a study basis.

### 2.2. Monitoring and Support

There are several studies and solutions devoted to monitoring the activity of the elderly with some motor independence and focused on keeping their caregivers informed. In the study presented in [14], the authors developed a system to monitor elderly people with cognitive disabilities. The data are collected by several means, such as sensors, wearable technology and beacons. The information is then processed on the elderly person's personal smartphone. Based on the information collected, the routines of the elderly are identified and the caregivers are alerted if the systems detect some deviation from the elderly's usual routines.

In the field of forensic analysis, the authors [15] propose the use of smartbands. Xiaomi Mi Band 2 and Fitbit Alta HR are used to obtain heart rate data so that it can be analysed to verify the veracity of the answers provided by the suspects.

The work [16] presents a platform that provides a set of services directed to the elderly population. The authors of the work indicate the advantages, the scalability due to the ease of adding new services and an easy interaction between the end users and the services. The services deal with the physical aspects of the elderly, monitoring their basic vital signs, their

normal environment and their daily routines. This monitoring reassures family members and caregivers, who are notified according to predefined policy alerts. The elderly develops feelings of security due to this assistance.

In [17], the authors propose a solution for supporting the daily tasks of the elderly. The developed system uses IoT devices and xBeacon transmission techniques combined with a hybrid algorithm in order to monitor the elderly's movements and to locate items that they may have introducing the feeling of anxiety and discomfort caused by the inability to find the lost items and potentially improving their quality of life. As with the works presented in [17,18], this is an intrusive solution, given that it requires a smartphone to obtain data from the xBeacon sensors.

The work presented in [19] proposes the implementation of an intelligent IoT system that, after being installed in the an elderly's home, enables a discreet and non-evasive monitoring of the elderly's wellbeing. The system obtains the necessary data from sensors to learn the elderly's behaviors his/her day-to-day life. The detected activities are reported to caregivers by utilizing an Android application so that they can make informed decisions regarding the safety and well-being of the elderly. The system consists of three components: IR sensors to detect the movements of the elderly; magnetic sensors to determine when the elderly leaves the house; and a controller that receives data and sends a message to caregivers if no movement is detected in a period of 45 min. This solution is essentially focused on collecting information on the elderly's home environment; however, it does not obtain important data related to the health of the elderly or data that would complement the system's information, such as heart beat which may sometimes generate false positives.

The authors of [20] propose a low-cost IoT solution for automation and care in the elderly's home. The proposed architecture allows the implementation of the following features: registration and analysis of daily routines; alert systems through emails or text messages to caregivers or social agents whenever an unusual situation occurs or when the elderly person is in danger (while providing real-time audio and video); control of comfort features, such as automatic lighting and temperature/humidity control; security and surveillance services; detection of possible hazardous situations such as fires, floods and the presence of toxic gases; real-time monitoring of the elderly through heart rate and oxygenation; emergency calls (through the use of a panic button). The prototype of the proposed solution, despite implementing most of the expected functionalities, does not implement the monitoring of vital signs or physical activity of the elderly.

Reference [21] proposed a localization system for AAL by using a WSN (Wireless Sensor Network) prepared to interact with IoT devices. The whole system is designed to be a low-cost solution, which is an essential requirement to allow the installation of a large number of sensors in the real environment. All the components used have the property of having low energy consumption. The tests permitted the conclusion that it is possible to accurately obtain information about the division in which the elderly is located.

The PreventIT System in [22] uses mobile devices as its front-end technology and more specifically with respect to smartphones and smartwatches for the collection of user data and the identification of possible risk factors based on the users' behavior. For the back-end technology, a cloud-based solution is used for protecting, processing and storing the personal data collected. The system has the peculiarity of considering the fact that the smartphone is not always carried by its users and behavioral information is also recorded and processed on smartwatches and subsequently transferred to the smartphone. Regarding the criteria defined in the present work, it does not allow the collection of data from the users' environment and it is considered an intrusive solution since it uses smartphones.

The FrailSafe framework [23] utilises short and medium terms in order to later create recommendation services and to avoid those risk conditions. This solution collects data based on human motion identification, through different data sources such as GPS systems, motion sensors or smartphones. The data obtained allows the improvement of the accuracy of the behaviours identification system and the prediction of risk conditions. As with the

previously mentioned work, this is also considered an intrusive solution because one of the data sources will be the smartphone. The non-monitoring of user environment data is also considered a limitation.

The main objectives of the Nestore project presented in [24] are the promotion of healthy aging and the improvement of the well-being of the elderly. For these objectives, an e-coach was developed that works in four different domains: physical activity, nutrition, social activity and cognitive activity. By using an intelligent system, which collects data from the daily activities of the user, and based on the analysis of the same data in the cloud through logical decision support systems, it will provide the elderly with tips and suggestions. The interaction with the elderly is performed by using a virtual assistant embedded in a chat application for smartphones and a tangible object that interacts through voice commands. The Nestore system switches between the two interfaces according to user preferences and context information, such as user proximity to the tangible object.

The eWALL project [25] uses a wall of the user's house and embeds an interactive device that will provide a variety of intelligent support services. These are grouped into the following categories: risk management; home security; eHealth; and healthy lifestyle management. In addition to supporting the users in their daily tasks, it provides notifications to caregivers whenever necessary. This solution monitors, in real time, the health and activity of the elderlies within their home. Despite presenting itself as a complete solution, it does not allow the real time location of the users inside their home to be obtained.

The authors of [26] propose a domestic low-cost system to support the independent life of senior users, especially for the elderly who suffer from some form of dementia. In order to obtain data from the environment and the users, several sensors are employed which include the following: Passive Infrared Sensors (PIR sensors); sound sensors; temperature and humidity sensors; sensors to detect the presence in bed; light sensors; and gas sensors. In this work, the authors also propose an algorithm for the Intelligent System of Notification and Alarm for the formal or informal caregiver in the case that any critical or potentially dangerous behavior is detected. This proposal is quite complete in terms of monitoring the elderly's home; however, it does not take into account vital data and the physical activity of the user.

The NOAH project [27] is an AAL solution that aims to provide care to independent seniors; this project does not, however, aim to implement an automated support tool, but rather aims to integrate and complement existing care practices of caregivers. At the architectural level, cloud services are responsible for collecting and processing data from the heterogeneous sensor network and from user interaction. At the application level, two mobile applications are available: the NOAHCare application to be used by caregivers, which allows monitoring the status of the elderly, the status of the housing sensors and the reception of alerts as well as visualization of statistics for various time periods; and the NOAH application to be used by the elderly themselves, which was developed to notify end users of various alerts such as a forgotten open door. It is also possible to establish telephone calls between the elderly and the caregivers. In view of the defined criteria, the proposed solution presents the fact of not monitoring the elderly's vital data, physical activity and position within the home as a limitation.

The CARE in [28] aims to create an intelligent environment to detect critical situations for the elderly users, such as falls. Thus, based on the data collected, the system signals possible situations that endanger the health of the elderly by intelligently creating and emitting alarms. This project uses only stationary technologies and not data from wearable devices.

The HABITAT project from [29] is an AAL platform that integrates different technologies, such as RFID, wearable devices, WSN and artificial intelligence systems. Its main objective is to create and integrate intelligent objects that are part of the daily life of the senior population, reducing health spending by limiting the need for personal assistance and enabling a better quality of life. This solution provides the following features: indoor

localization through wearable RFID tags; correct posture monitoring when the elderly person is seated; motion monitoring by using wearable belts (with built-in actigraph sensors) and wall panels; and mobile devices for user interaction. Despite presenting itself as a very complete solution for monitoring the physical activity of the elderly, it is considered an intrusive solution and it does not take into account the need to monitor the house or the user's vital data.

The authors of [30] propose a solution that allows collecting data related to the physical activity of the elderly, such as heartbeat, GPS location and number of steps taken. The system consists of three components: smartwatch application, smartphone application and a back-end with webservice. The smartwatch app is intended to continuously collect the user's heart rate, GPS location and number of steps for subsequent transmission to the smartphone application. The smartphone application aims to receive smartwatch data and, when there is connectivity to the server, to synchronise the data through a webservice. The backend has two main objectives: To provide a database that allows storing the data collected by the application and to provide features using the REST architecture to facilitate the operations of querying, creating and updating the information in the database.

### 2.3. Activity Recognition

The work proposed by the authors in [18] suggests a solution for the recognition of some daily activities of the elderly: drinking, washing hands, urination and defecation.

The data are collected from the environment and from the elderly person by using sensors: accelerometers, gyroscopes and Bluetooth Low Energy (BLE) devices. The data are sent to a smartphone that acts as a gateway for communication with the server. The server then uses the data and applies some decision algorithms, such as J48, RepTree, SMO (Sequential Minimal Optimization) and RF (Random Forest) among others, to recognize the activities which if identified are sent to the elderly's caregiver for information.

In [31], a low-cost solution for the recognition of human activity is presented. This system uses a Microsoft Kinect V2 sensor for RGB-D data acquisition and the Naive Bayes (NB), Multi-Layer Perceptron (MLP) and Random Forest (RF) learning algorithms for the classification of different types of activities. Although this work does not address most of the challenges posed to AAL systems, it presents the advantage of implementing an economic solution which permits the recognition of the tasks that the elderly are performing.

## 3. Architecture of the Testbed System

In order to evaluate the usability of smartbands by senior users, an architecture was designed for a service aimed for research and to provide an answer to the problem set out in Section 1 regarding the acceptance of the use of a smartband by senior users in their daily activities. In this section the architecture is discussed since it serves as the basis for the implementation of the solution's testbed.

As presented in Figure 1, the architecture relies on the client-server model and consists of four modules:

- The sensing module is comprised of the devices, including smartbands, which acquires health and physical activity data from the elderly.
- In the data centralization module, the Gateway device is a central and unique device located in the home, which is responsible for receiving information from all smartbands present in the home and sending it to the management and analysis module by using the cloud.
- The management and analysis modules are responsible for receiving, storing and analysing the data. The analysis detects any anomalous behaviour and sends a notification to the caregiver through the information and alert module. This module also manages information regarding caregivers, homes and the elderly.
- The information and alert modules are responsible for providing the caregivers with information about the physical activity and health status of the elderly. In cases of

detection of behaviours or parameters outside of specific thresholds, this module notifies the caregiver.

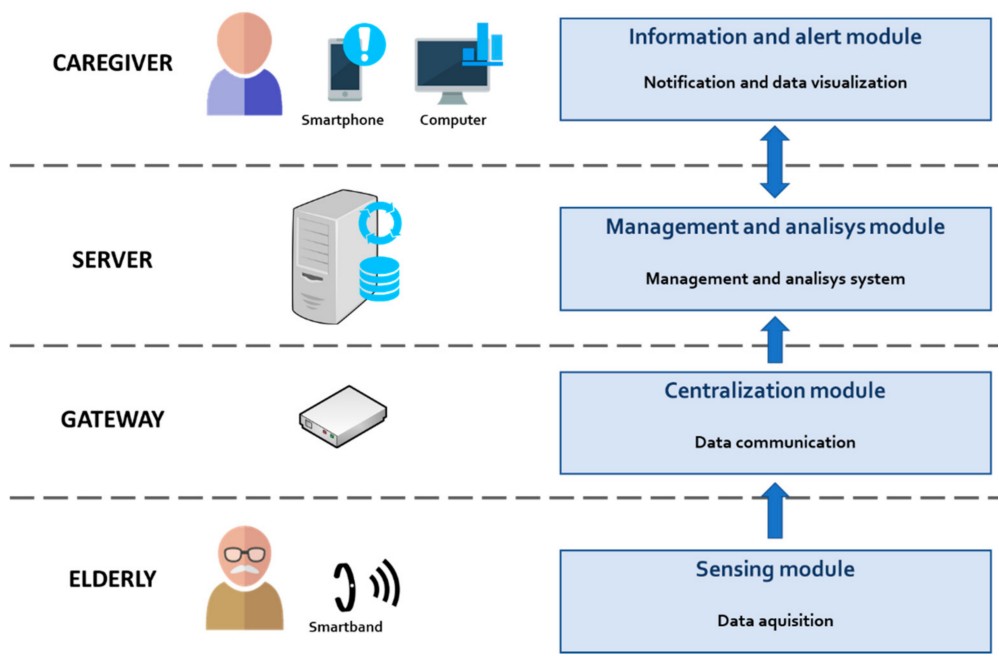

**Figure 1.** Testbed architecture.

*3.1. Sensing Module*

As presented in Figure 2, the Sensing Module is comprised of two elements: the elderly and a smartband: (i) The elderly, who is the target to be monitored wears (ii) a smartband on his/her wrist to collect data, including heart rate per minute (bpm), calories burned, steps and daily meters made; (iii) the data obtained by the smartband are sent to the gateway via a Bluetooth Low Energy connection.

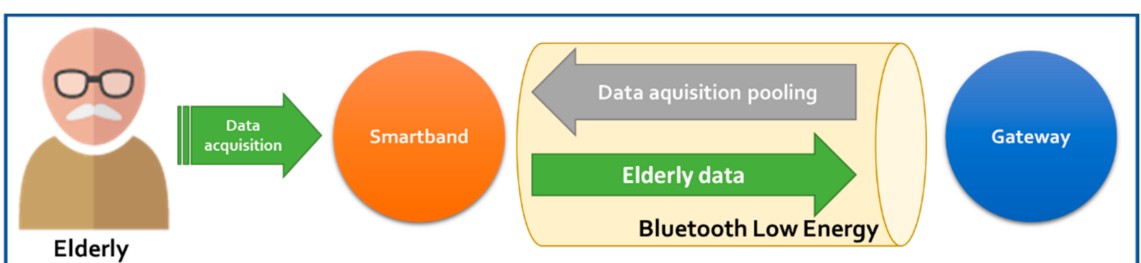

**Figure 2.** Sensing module.

*3.2. Centralisation Module*

As shown in Figure 3, there may be several smartbands in the house for monitoring different elderly people. The gateway device periodically requests the elderly person's data from each smartband, obtains the answers and transmits the obtained data to the Management and Analysis Module through a Wi-Fi or Ethernet network. Communication between gateway and smartband is connection-oriented, meaning that the Bluetooth communication stack will handle all the details in order to provide a reliable communication channel. The gateway is physically located in the elderly person's home and is configured to periodically send the elderly person's data to the platform server.

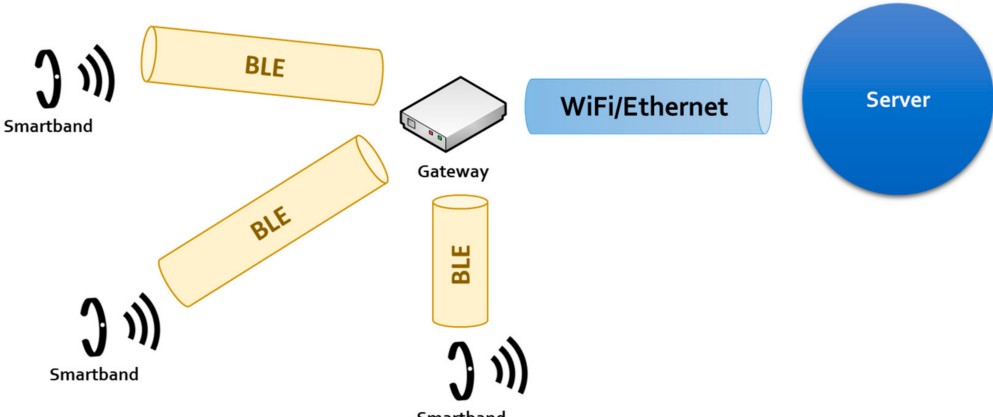

**Figure 3.** Centralisation module.

### 3.3. Management and Analysis Module

The management and analysis module, represented in Figure 4, receives and processes the data sent by the sensory module. The processing is executed by a software Processing Agent, which during the data analysis may perform queries relative to the information stored in the database. For instance, the Process Agent may compare the data regarding the physical activity of the elderly with previous datasets and detect that the elderly individual is becoming more inactive. In this particular case, the Notification Agent sends an alert to the caregiver through the web platform. After the end of the analysis process, the processed information is stored in the database. This module also provides a web portal, through which the caregiver can monitor the elderly person and browse his/her daily information. In order to meet the requests coming from the web platform or API (Application Programming Interface), the Query Agent is utilized, which can return statistical and monitoring information of the elderly.

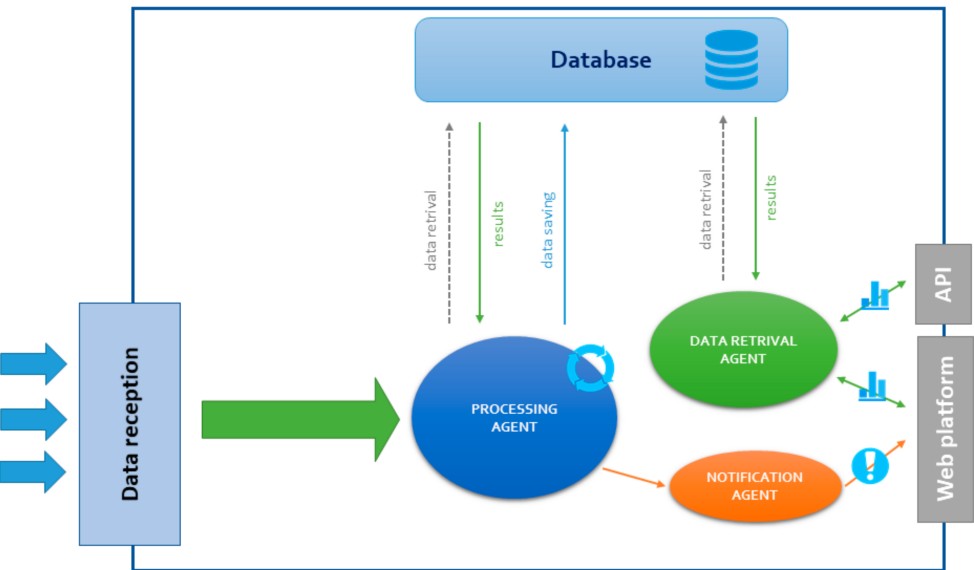

**Figure 4.** Management and analysis module.

### 3.4. Information and Alerts Module

The Information and Alerts Module, illustrated in Figure 5, provides the user interface for the caregiver to access information about the physical activity and health of the elderly person.

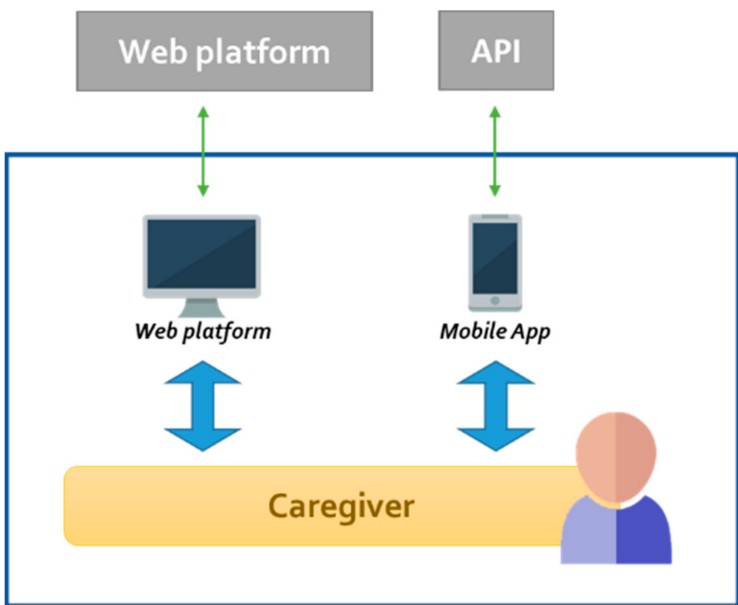

**Figure 5.** Information and alerts module.

There are two methods of access:

- Through a web portal that is provided by the Management and Analysis Module that can be accessed via a browser;
- Through a mobile application that is installed on the caregiver smartphone, which communicates directly with the API of the Management and Analysis Module.

## 4. Prototype Implementation

This section presents the equipment used and describes how the prototype was implemented, according to the reference architecture previously described.

### 4.1. Equipment

- Sensing module

In order to acquire the elderly's data, the Xiaomi Mi Band 2 (Technical specifications available at: https://www.mi.com/global/miband2, accessed on 14 January 2021) smartband was used, as illustrated in Figure 6. This bracelet device can monitor movements (steps, metres and calories expended) and heartbeat, as well as display information such as the time. The Xiaomi Mi Band 2 possesses IP67 protection certification and, thus, it is resistant to dust and water immersion of at least 1 metre deep for 30 min. Another relevant feature is that it is made of malleable and anti-sweat silicone which provides greater comfort in its use.

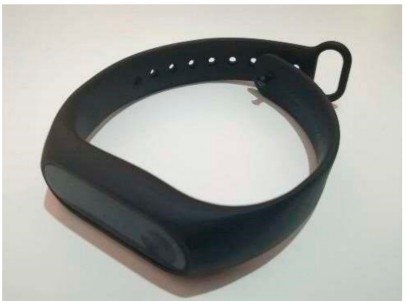

**Figure 6.** Xiaomi Mi Band 2.

The Xiaomi Mi Band 2 smartband carries a built-in photoelectric sensor for measuring heart rate, as displayed in Figure 7.

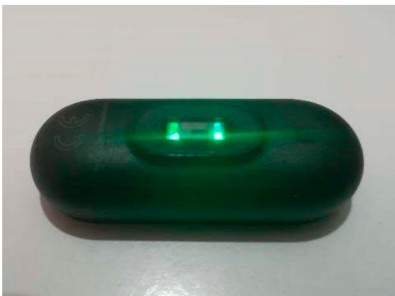

**Figure 7.** Photoelectric sensor.

The Mi Band 2 and bracelet set are very light and weighs approximately 17 g. It possesses a battery capacity of 70 mAh and autonomy of about 20 days. It is also possible to communicate with this device via Bluetooth Low Energy (BLE) version 4.0.

Based on rapid prototyping platforms, which allow the implementation of hardware projects with software for so-called embedded systems, the Raspberry Pi Zero W (Technical specifications available at: https://www.raspberrypi.org/products/raspberry-pizero-w (accessed on 15 January 2021) IoT device was chosen, as shown in Figure 8, because this equipment has all the necessary features such as reduced dimensions (approximately half the size of a credit card), BLE communication (useful for communication with the smartband), processing capacity of 1 GHz (single core) with 512 MB of RAM memory and internet connectivity via Wi-Fi or LAN. In terms of energy, only a 5 V 2.1 A power supply is required.

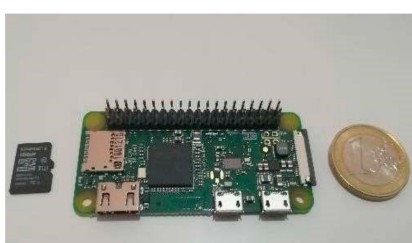

**Figure 8.** Raspberry Pi Zero W.

Taking into account the scope of the project, another positive feature is its low cost: Its cost is around €21 + VAT (including 5 V 2.1 A power supply and 16 GB micro SD card).

In terms of software, this equipment runs on an operating system based on Linux called "Raspberry Pi OS" (Operating system documentation avaiable at: https://www.raspberrypi.org/documentation/raspbian/ (accessed on 2 February 2021). This operating system is free and based on Debian, which is optimized for the Raspberry PI hardware.

- Management and analysis module.

For the server equipment, which houses the software platform, a computer with the following characteristics was used: Intel Core i7 processor with 4 cores at 2.4 GHz, 8 GB RAM memory and 1 TB hard disk.

- Information and alerts module.

In this prototype, only the web platform of the service was implemented, which does not require much processing capacity for the device used by the caregiver. As it is only necessary to access the web platform via a browser to view the data processed by the platform, the caregiver was provided with a laptop computer.

### 4.2. Operation

When the gateway begins, as presented in Figure 9, it loads the settings stored into memory in the settings.conf file. This file contains the following settings: (i) the smartbands MAC address list, in which each address corresponds to a device and consequently to a distinct elder person; and (ii) the timer, which corresponds to the waiting time between pooling data acquisition from the smartbands. Next, a scan is made of the BLE devices that are within the gateway's range. If the detected devices match some of the defined MAC addresses, then the gateway device communicates with the smartband via BLE by pooling it to perform an acquisition of the elderly's data, as displayed in Figure 9. After reception, the data are pre-processed and transmitted to the server.

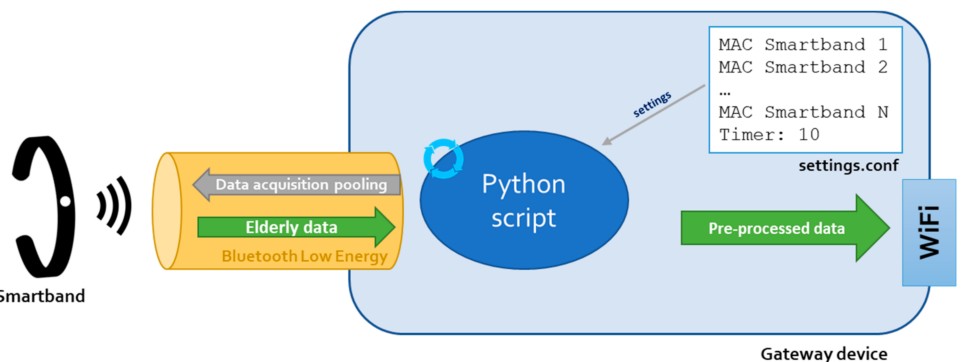

**Figure 9.** Gateway device operating mode.

The gateway then waits to resume the pooling process again. Figure 10 illustrates the gateway operation algorithm.

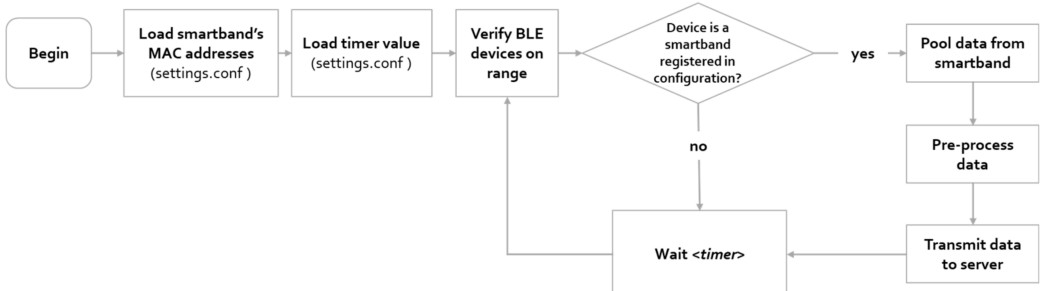

**Figure 10.** Operation algorithm of the gateway script.

As defined in the platform's architecture, the data sent by the gateway are received by the server (Management and Analysis Module), which processes and stores it in the database. The server provides a web platform that allows access to two types of users: the administrators and the caregivers.

- Administrator user

Users with an administrator profile can manage (create, read, update, and delete) the information of the elderly, e.g., home location, caregiver detail, etc. Figures 11 and 12 illustrate the platform's pages available for managing the elderly's information.

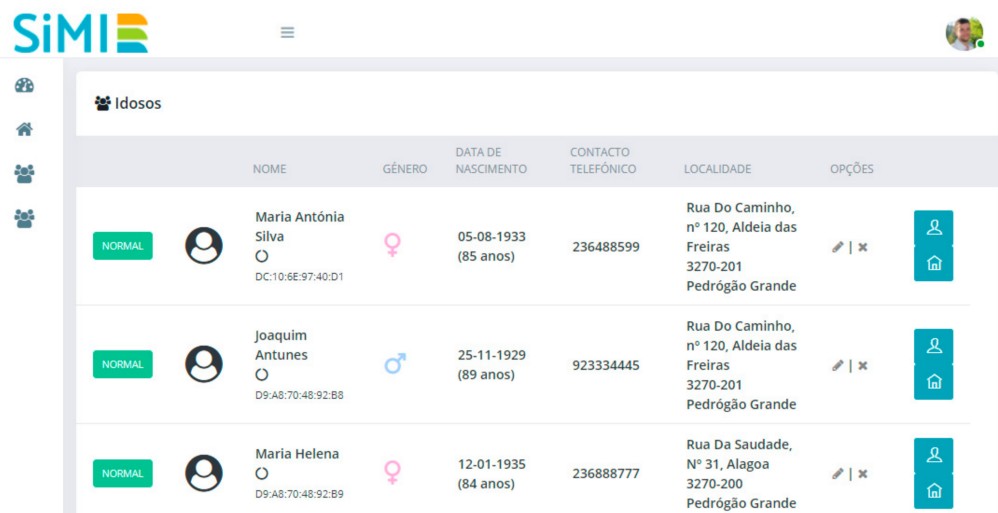

**Figure 11.** The elderly's management page.

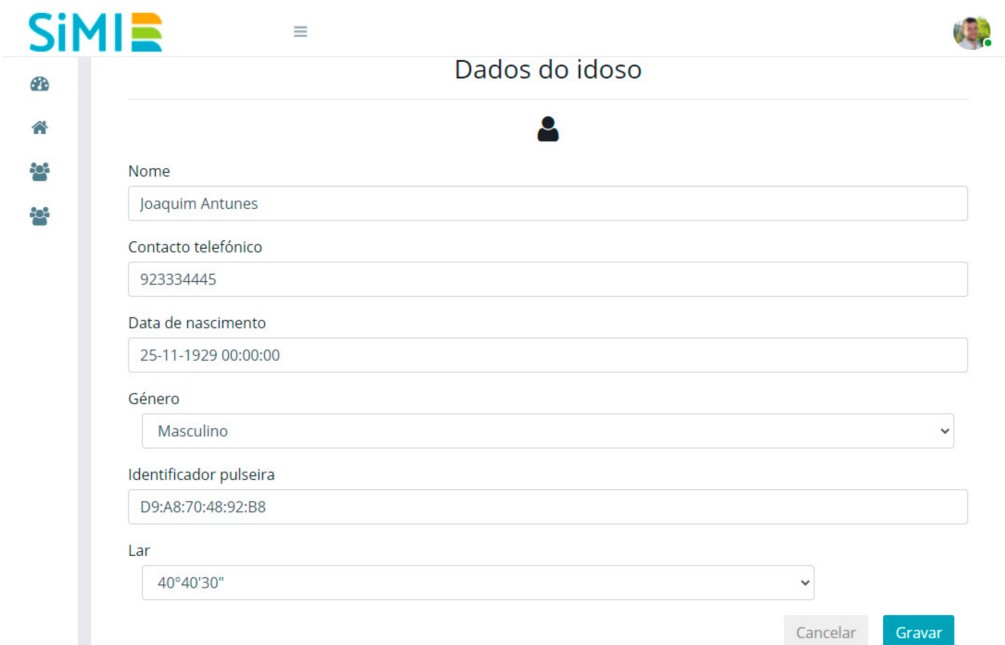

**Figure 12.** The elderly's editing page.

- Caregiver user

Users with a caregiver profile can access information about the elderly's activity, e.g., heart rate, steps walked and distance (metres) walked. Figure 13 illustrates the home page for the caregiver for viewing the latest measurement data.

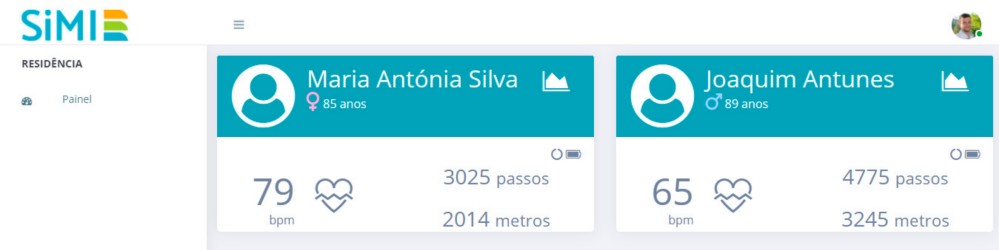

**Figure 13.** Web platform with the elderly's data.

The system also provides statistical data regarding the elderly's physical activity and cardiac monitoring for the last 24 h, as displayed in Figure 14.

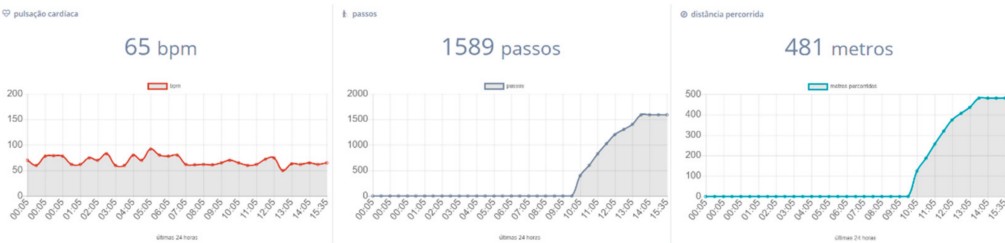

**Figure 14.** The elderly's physical activity statistics.

## 5. Evaluation of the Proposed Solution

### 5.1. Test Scenario

In order to perform tests on the developed prototype and usability tests on the smartband, it was necessary to define a test scenario that was as similar as possible to the actual usage environment of the solution, as illustrated in Figure 15.

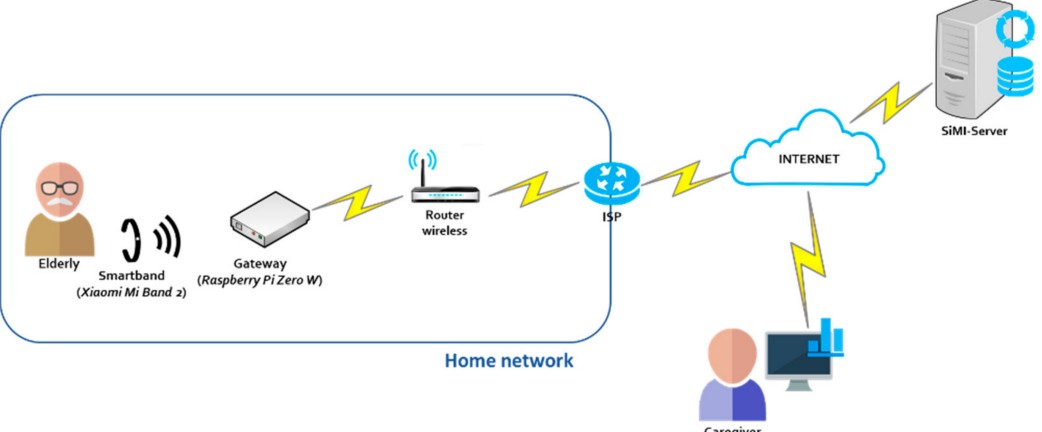

**Figure 15.** Test scenario.

This scenario was implemented in the elderly's home. Users have to put on the smartband and go about conducting their daily routines. As this is a non-intrusive solution, there was no need to interact with the device.

There was no problem in terms of the energy autonomy of the smartband since it was loaded before the beginning of each test. Bearing in mind that each test lasted 5 days and that the device's autonomy is about 20 days, there was no need to charge during the test period. It should be noted that, in the normal use of the solution, the caregiver would be responsible for charging the smartband.

The following equipment was used: a smartband Xiaomi Mi Band 2; a Raspberry Pi Zero W (Gateway); a wireless router with internet connection; a computer with server functions; and a laptop computer for the caregivers to access the web platform.

### 5.2. Usability Tests of the Smartband

In order to assess the level of acceptance of the usage of smartbands by elderly users, a set of tests was carried out with 11 elderly people. The participants' ages ranged between 65 and 78 years old and the average age was 68.9. Regarding gender, 63.6% were female and 36.4% were male.

A Wi-Fi network with internet connection and a gateway device (Raspberry Pi Zero W) were properly configured and installed in each elderly person's home and a smartband (Xiaomi Mi Band 2) was provided to each elderly person.

The caregivers of the participants were trained to access the web platform and to perform the necessary actions to retrieve information.

Each test lasted for 5 days. All the elderly participants were asked to wear their smartband, as illustrated in Figure 16, at all times during this period.

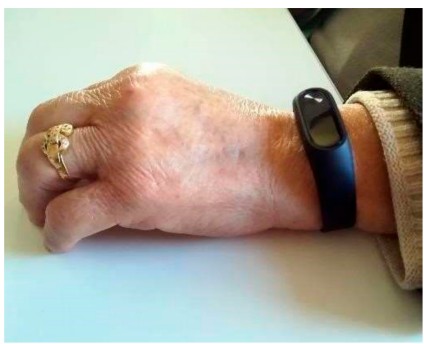

**Figure 16.** Elderly person wearing a smartband.

At the end of each test, the level of satisfaction of the elderly participants with the use of the smartband was assessed by using an interview with the following questions.

1—Did you feel any discomfort while using the smartband? If yes, in which situations?
2—Did you feel any pain while using the smartband? If yes, in which situations?
3—How do you rate the weight of the smartband?
4—Did you feel that the smartband is bothersome?
5—Did you remove the bracelet during the test?
6—In the future, would you consider wearing the smartband in your daily life?
After analysing the answers, the following was found:

- None of the respondents reported feeling pain while using the device;
- No discomfort was revealed in 90.9% of the elderly when using this device, while the remaining 9.1% reported feeling discomfort;
- Regarding the weight of the bracelet, 54.6% stated that it was very light, 36.4% stated that it was light and only 9% indicated that it was a not very light device. Only 9.1% of the elderly considered it as an uncomfortable object;
- During use, 18.2% said they had removed their smartband to perform tasks such as taking a bath or washing up because they were afraid of damaging the device with water;
- On the last question, 91% of the users stated that they would consider using the device in their daily lives.

These results suggest that the vast majority of users did not feel that the smartband was intrusive and that they would even consider using it in their daily lives.

### 5.3. Usability Tests of the Web Platform

In order to assess the caregivers' level of satisfaction in using the web platform, a questionnaire was conducted to evaluate their level of satisfaction. The caregivers of the elderly who performed the test presented in Section 5.2 were involved in this activity. At the end of the test for each elderly person, the respective caregiver was interviewed with the following questions.

1—How difficult is it to access the data of the elderly person's heart rate and physical activity (steps taken and metres walked)?
2—How difficult is it to interpret the statistical information presented?
3—Do you think the platform is intuitive?
4—On average, how many times a day did you access the platform?
5—In the future, would you consider using this service again to monitor the elderly person in your care?
After analysing the answers, the following was found:

- In accessing the elderly's heart rate and physical activity data, 81.8% had no difficulty; on the other hand, 18.2% reported having experienced a bit of difficulty.
- In interpreting the statistical information presented by the web platform, 45.4% had no difficulty, 27.3% had a bit of difficulty and 27.3% had some difficulty.
- 72.8% of the caregivers think that the platform is very intuitive, 18.2% indicated that it is reasonably intuitive and 9% consider it not to be very intuitive.
- It was also found that caregivers accessed the platform 3.2 times a day, on average.
- All caregivers considered using this service to monitor the older person in their care in the future.

*5.4. Discussion*

Not being aware of other similar projects, we will compare the results obtained and those described in Sections 5.2 and 5.3, with the works presented in refs. [11,12].

The authors of [11] intend to explore usability issues as well as possible limitations in the use of smartwatches. To this end, tests were carried out with 30 users aged between 20 and 43 for a week, in which users had to perform various tasks using the mobile device. Based on the results obtained, the authors identified some usability problems caused by smartwatches in both the interaction with the device and in localised discomfort and pain caused by smartwatches, in part due to the size and weight of the devices. Despite the work presented in [11] having as their objective the analysis of the usability issues of smartwatches and even though relevant conclusions were attained, the authors did not carry out tests with elderly people. On the other hand, the object of the study is limited to smartwatches that, due to their dimensions and weight, caused pain in some participants and an analysis of the use of smartbands was not contemplated. Considering the criteria defined in our work and given the non-intrusive nature of the solution, the interaction between the user and the device is not an object of analysis; in our proposed solution, a "use and forget" approach is adopted in which the elderly person only has to wear the bracelet, with the only additional interaction being the caregiver's act of charging the smartband when needed.

The work presented in [12] aims to investigate the motivations for the use of physical activity monitoring technologies by the elderly. In order to obtain the answer to the research question, the authors interviewed 1013 users over 50 years of age residing in Switzerland by telephone. The authors concluded that younger elderly males have a greater interest in monitoring technology; however, they show a potential acceptance of physical activity monitoring devices by older elderly participants. However, for this group of elderly people, it is mentioned that there is a need for further studies on the motivational and usability aspects related to the use of mobile devices. In order to obtain the results, the authors did not implement or use a prototype or solution; the participating users were also not subjected to any type of tests and only had the prior knowledge of each user being considered. With our solution, a data acquisition prototype was developed that was used in a sample of an elderly population and, by using surveys, we have concluded the following: (i) the smartband did not cause pain in its use; (ii) only 9.1% felt discomfort when using it; (iii) it is considered a light device that does not disturb the individual when performing most daily tasks; (iv) most seniors surveyed are willing to use this device in their daily lives.

In addition to the differences mentioned above, our work presents important conclusions for the area of Ambient Intelligence in general and of AAL, in particular, since it was possible to conclude that the majority of the elderly did not consider the use of smartband to be intrusive or that it constituted an impediment for carrying out their daily tasks. It was also concluded that this type of wearable device is viable for monitoring physical activity, as it is well accepted by senior users. On the other hand, caregivers found the proposed platform very intuitive and considered the possibility of using a monitoring service for the elderly in the future.

## 6. Conclusions and Future Work

The primary goal of the study described in this paper was to answer the following research question: "What is the level of acceptance, by the elderly population, of the use of smartbands to monitor their daily activities?" In order to answer this question, the following objectives were defined: (i) analysis of monitoring systems for the elderly population; (ii) proposal of an architecture using the IoT paradigm in the context of AAL systems; (iii) implementation of a prototype; (iv) tests with elderly people and caregivers in a real environment; (v) questionnaires to participants in the test session and analysis of the results obtained.

According to the context presented in Section 1, with respect to the population ageing and their isolation combined with the emergence of the IoT technologies, this study's main objective was to assess the level of acceptance of the use of a smartband by senior users in their daily activities. To this end, a testbed architecture was designed, using smartbands to monitor physical activity and to collect data on the health of the elderly.

Based on the architecture, a prototype was implemented using the following: (i) the Xiaomi Mi Band 2 smartband for movement monitoring and heartbeat measurement; (ii) a Raspberry Pi Zero W IoT device as a gateway device; (iii) a software agent to analyse the data; (iv) a web portal for the caregivers to access information about the elderly's activities.

A set of tests was conducted with 11 elderly for 5 days, after which the elderly and their caregivers were interviewed to collect data about the tests. Based on the data from the interviews, it is possible to conclude that most users did not feel that the smartband was intrusive to their daily tasks and even considered using it in the future. In addition, 72.8% of the caregivers considered that the platform was very intuitive, having accessed it on average 3.2 times a day to view their elderly person's status. All caregivers considered that they would use this service in the future to monitor the elderly.

The results obtained showed that the system as designed and prototyped can be a good starting point for the development of a production solution that addresses the problem in a similar manner. It was also concluded that the use of smartbands as monitoring devices is feasible, as they are well accepted by senior users.

The main contributions of our study are listed as follows: (i) An analysis of the feasibility of using smartbands for monitoring the elderly population; and (ii) the evaluation of the acceptance of caregivers in relation to an elderly monitoring system.

Even though the prototype solution developed met the basic requirements of the proposed architecture, some aspects were identified that may be improved in the future. The development of a mobile application which would allow the caregiver easier access to the platform data would be helpful, as well as the implementation of an alert system to actively notify the caregiver whenever the system detects changes in the elderly person's behaviours.

**Author Contributions:** Conceptualization, L.C., D.F. and A.P.; data curation, L.C. and D.F.; formal analysis, A.R., C.R., J.B. and A.P.; funding acquisition, A.R., J.B., C.R. and A.P.; investigation, L.C. and D.F.; methodology, A.P.; resources, A.R., C.R., J.B., J.R., N.C. and A.P.; software, L.C. and D.F.; supervision, A.R., J.B., N.C. and A.P.; validation, N.C., A.R., J.R., C.R., J.B. and A.P.; writing—original draft, L.C., D.F., N.C., A.R. and A.P.; writing—review and editing, L.C., D.F., J.R., A.R., C.R., J.B. and A.P. All authors have read and agreed to the published version of the manuscript.

**Funding:** This work is financed by National Funds through the Portuguese funding agency, FCT-Fundação para a Ciência e a Tecnologia within project UIDB/04524/2020 and was partially supported by Portuguese National funds through FITEC-Programa Interface with reference CIT "INOV-INESC Inovação-Financiamento Base" and by Portuguese Fundação para a Ciência e a Tecnologia-FCT, I.P., under the project UIDB/50014/2020.

**Institutional Review Board Statement:** Ethical review and approval were waived for this study, due to it was not a health study and no threat to the health and life of the participants occurred. Furthermore, to ensure the privacy of participants, all the information of users' answers was anonymized and de-identified before proceeding with its analysis.

**Informed Consent Statement:** Informed consent was obtained from all participants involved in the study.

**Acknowledgments:** The authors acknowledge the Computer Science and Communication Research Center for the facilities granted in the implementation of part of this work in the context of the Smart IoT Ecosystems research line and the Mobile Computing Laboratory of the School of Technology and Management of the Polytechnic of Leiria.

**Conflicts of Interest:** The authors declare no conflict of interest.

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
