# Peer review of "Usability of Smartbands by the Elderly Population in the Context of Ambient Assisted Living Applications"

_electronics, doi:10.3390/electronics10141617_

Round 1

Reviewer 1 Report

The topic of this study is very interesting. However, there are some aspects that need major improvements. 

    1. There is a Critical need to discuss the findings. The authors may compare the findings of this study with those in the related literature.
    2. It is expected that practical contributions to be clearly stated.       

    3. The abstract section should highlight the novelty of this study findings and contributions. 

    4. A more creative academic writing is encouraged.

    5. Detailed information regarding the context, procedures, and participants of this study are required.

Reviewer 2 Report

After reading this manuscript, I recommend this article can be accepted after a minor revision. Implementing this revision is easy. Thus 1. The authors employed a Raspberry Pi Zero device as the gateway device. I suggest the authors can describe the operating system of this device; thus, readers of this article can understand it is a computer for sending monitored data to a server. 2. Did those elderly participants have the responsibility of recharging the Xiaomi Mi Band 2 device during the 5 days? Or someone recharged this device. The authors may add similar information to show that the Xiaomi Mi Band 2 device is suitable for monitoring elderly persons.

Reviewer 3 Report

1. The structure of the related work is blur. The introduced papers are not classified or described in certain pattern. 2. The test result and comparison on certain parameters (execution time, packet loss rate etc) have not been presented. Thorough performance analysis is required. 3. The novelty of this paper is not enough, It seems that the proposed method does not have much academic contribution but just small adjustment on exising application.

Reviewer 4 Report

The paper focused to the real-time collection of biometric data in humans. This paper proposes assessing the level of acceptance of the usage of a smart band by senior users in their daily activities.

Several elements need to be corrected in the paper:

  • A very simple description of the testbed architecture is present in the paper in section 3. It should be more detailed in order to give to the reader a more precise overview of the prototype realized and tested by the authors.
  • Please improve the quality of the figures (too small and quite illegible).
  • You can compare the obtained results for the proposed architecture with other publications containing similar architectures.

Round 2

Reviewer 3 Report

It seems that the authors have improved the quality of the manuscript during revision. I recommend accepted now.